

# Lagrangian aerosol particle trajectories in a cloud free marine atmospheric boundary layer: Implications for sampling

Hyungwon John Park[1#], Jeffrey S. Reid[2*], Peter F. Caffrey[3], Maria J. Chinita[4], David Richter[5]

[1]NRC, US Naval Research Laboratory, Monterey, CA
[2]US Naval Research Laboratory, Monterey, CA
[3]US Naval Research Laboratory, Washington DC
[4]Joint Institute for Regional Earth System Science and Engineering, University of California Los Angeles, Los Angeles, California, USA
[5]University of Notre Dame South Bend, IN
[#]Now at Honeywell FM&T, Kansas City, MO

*Correspondence to*: Jeffrey S. Reid (jeffrey.s.reid20.civ@us.navy.mil)

**Abstract.** Meteorological processes such as gust fronts, roll structures, internal boundary layer development, and smaller scale turbulence complicate the physical interpretation of measured aerosol particle properties, fluxes, and transport in the marine atmospheric boundary layer (MABL). To better decipher maritime aerosol measurements by aircraft, ships, and towers we describe an ensemble of particle trajectories using high resolution large eddy simulations (LES) of surface-emitted aerosol particle within a Lagrangian framework. We identified two clusters of particle trajectory types from which we created probabilistic distributions of particle histories: a) short lived particles that do not exit the surface layer and are subsequently deposited back to the ocean; and b) much older particles that are able to exit the surface layer into the mixed layer and subsequently oscillate up and down through convective roll structures. After emission in a neutral atmosphere, particles slowly disperse through the MABL requiring, on average, up to 100 minutes to mix to the ~570 m deep mixed layer inversion. However, for even slightly unstable conditions, particles are rapidly transported to the top of the MABL in roll structure updrafts, where they then more slowly diffuse downwards, with some similarities to a looping plume rise to the stable inversion followed by fumigation. Consequently, particles can exhibit a bimodal lifetime distribution that results in different particle ages by altitude. Further, based on wind speed and stability, the initial looping behavior following an emission event spans 15 to 30 minutes and may result in sampling "blind spots" up to 15 km downwind. Overall, our findings suggest that there should be a consideration of the representativeness of particle ages, even in what is often assumed to be a well-mixed MABL. This representativeness is related to how long particles have been suspended and whether they were sourced locally, which is critical for situations such as for measuring wind generated emissions or ship track plumes. Further, the Lagrangian technique for treating the particle transport captures the inherently random motion of the MABL turbulence and does not exhibit artificial numerical diffusion. As such, it produces differences when compared to a traditional, column-based eddy-diffusivity approach used in mesoscale to global scale models. We used the LES to drive a 1D column model to approximate single grid point physics. The results were starkly different near the surface, with the 1 D column model missing the looping behavior and showing a slow upward dispersion. This difference in the 1D and LES



frameworks is an excellent example of sub-grid problems and may explain some of the differences between observations and global and meso-scale model simulations of marine particle vertical distribution and dry scavenging.

## 1 Introduction

Within the Planetary Boundary Layer (PBL), aerosol particles reside as fundamentally discrete entities that are transported across many spatial-temporal scales, ranging from larger scale synoptic and mesoscale flows to organized updrafts and downdrafts in and around clouds, to the smallest fine-scale turbulence adjacent to the ocean and land surface. Whether generated from sea spray (*Lewis and Schwartz*, 2004), aeolian processes (*Ginoux et al.,* 2001) or ship emissions (Durkee et al., 2000), primary aerosol particles act as vessels of mass, cause electro-magnetic, optical, and radiative responses (*Dave,* 1969; *Hulst and Hulst,* 1981), become cloud condensation nuclei for cloud droplets (*Albrecht,* 1989; *Dadashazar et al.,* 2017; *Dzieken et al.,* 2021), and at times for coarse and giant particles, momentum and energy exchange between the surface and the atmosphere (*Peng and Richter,* 2019; Veron 2015). Thus, characterizing particles' spatial-temporal distribution remains at the forefront of a large portion of the atmospheric research community. Indeed, given the variety of sources, transport, transformations, and sinks of atmospheric particles, the challenge of reconciling observations and models, as well as developing reliable parameterizations for aerosol fluxes, remains paramount.

The combined 2019 Office of Naval Research (ONR) Propagation of Interseasonal Tropical OscillatioNs (PISTON), the NASA Cloud, Aerosol, and Monsoon Processes Philippines Experiment (CAMP2Ex; *Reid et al.,* 2023), and the 2023-2024 ONR Moisture and Aerosol Gradients / Physics of Inversion Evolution (MAGPIE) field campaign were a natural progression of interagency studies that included investigations of evolving four-dimensional Marine Atmospheric Boundary Layer (MABL) structures and their relationships to the ocean surface and clouds. Given the difficulty of capturing the volumetric variability of the MABL to infer aerosol fluxes and physics, the missions needed to develop new strategies of collecting and projecting surface, ship, and airborne observations to models in an object-oriented manner (*Reid et al.,* 2023). In situ observations are by definition made at a single point in space or a trajectory across time (e.g., tower, aircraft, ship). At the same time, high-resolution models are challenged to deterministically represent such a point. Comparisons can be made between models and observations statistically, but the aliasing inherent with in situ sampling prevents convergence of measured fluxes to the environment even in idealized model environments (e.g., *Park et al.,* 2022). When we consider fine scale MABL features and variability in real world scenarios, such as along wind fronts and point sources such as ship emissions, the challenge of projecting observations onto models to infer physics such as fluxes, lifetime, and transformations becomes quite daunting.

To illustrate these observational challenges, consider the June 24, 2024 MAGPIE case at Barbados that represents a typical convective line propagating through the tropical/subtropical Atlantic Basin (Fig. 1). The visible GOES 16 ABI image (Fig.1(a)) taken just after at sunrise (9:50Z) and enhanced in brightness and contrast, captures a convective line propagating across Barbados and into the other Lesser Antilles (St Lucia and St. Vincent), ahead of a near surface jet associated with the



onset of a Saharan Air Layer (SAL event; Dunion and Velden, 2004; Tsamalis et al., 2013). Complementing this view, RadarSAT-2 synthetic aperture radar (SAR) data, collected over the region at approximately the same time and post-processed for wind speed (using CMOD4) at 250 m resolution shows the natural variability in wind features (Figure 1 (b)). This case shows a western ~5 m/s wind regime being overtaken by a north-south oriented gust front ahead of the convection over the Lesser Antilles, followed by a wider high wind speed regime to the east. While there are some small areas of

artifact from high amounts of cloud ice (marked), there is significant amounts of variability in wind patterns. Indeed, what often seems like uniform wind features to observers at a tower is in fact a spatially heterogeneous sample of structured wind and cloud features that often preferentially minimize variance in the streamwise (i.e., along wind) direction. Features with coupled cloud-wind relationships include low-wind orographic wakes, edge-induced channeling or jets, cold pools, etc. Most notable is the formation of individual wind streaks that are a result of roll structures that are 5-40 km along wind and ~2-4

km apart. These roll structures are particularly noticeable as cloud streets in the associated GOES-16 visible image associated with this SAR collection. Such features can significantly influence local sea spray production (e.g., Kapustin et al., 2012).

There is an inherent assumption that fundamental processes can be inferred from the instantaneous local state (*Smith, Park, and Consterdine* 1993; *Covert et al.* 1996; *Geever et al.,* 2005), and yet missions that track the gradients and evolution of the

MABL (e.g., *Reid et al.*, 2001; *Kapustin et al.,* 2012) and Figure 1 clearly demonstrate the many ways such assumptions can be violated. Figure 1 provides numerous questions, including: 1) If we sample at a point, how representative is that measured aerosol state of local conditions?; 2) How much is a sampled aerosol a result of nearby production or long range transport?; 3) Given presumable enhanced white capping along wind streaks, are there differences in the relative ages of aerosol particles measured near the surface on a ship or at a coastal site, versus what is measured on an aircraft at its lowest level

(say 30 or 100m) to the top of the marine boundary layer, ~400-700m? ; 4) What are the implications for the white cap production?; 5) If a local flux measurement using eddy correlation, vertical profile, or box methods (e.g., see *Lewis and Schwartz* 2004) is made, to what extent can that measurement be influenced by non-local forcing and at what ranges? These questions no doubt have complex and qualified answers. But a scale analysis would be useful to understand under what circumstances wind variability should be a concern to observations and observational objectives.

PISTON and CAMP[2]Ex focused on the North Tropical Western Pacific with both ship and P3 observations. Now, MAGPIE's sampling strategy is centered on long term monitoring at the University of Miami Barbados Atmospheric Composition Observatory (BACO, *Prospero et al., 2021*) tower at Ragged Point with aircraft observations just offshore. Comparisons can also be made to the Barbados Cloud Laboratory (BCL) 450 m to the southeast managed by the Max Planck Institute for Meteorology (*Stevens et al.,* 2016). With the above questions in mind, we must consider for all these missions

that ships and towers can sample near the ocean surface, but samples in a streamwise, Eulerian manner as the air mass moves past. Aircraft can sample in any direction to capture more structure, but cannot sample near the ocean surface and are embedded in larger and poorly observed meteorological structures. Remote sensing from the surface or aircraft can further





improve coverage, but also suffer their own sampling biases. Even comparisons between aircraft and ships, or say BACO and BCL only 450 m from each other, may also at times be problematic. Therefore, to better understand the sensitivity of

aerosol sampling to variability in wind features in fair weather tropical to subtropical conditions sampled during PISTON, CAMP²Ex, and now MAGPIE, a scale analysis was performed of aerosol particle trajectories and relative lifetimes using a series of Large Eddy Simulations (LES). To be sure, there have been numerous applications showing boundary layer plume dispersion. Further, full Lagrangian simulations of a domain and suitable resolution to replicate Figure 1 is a massive endeavor.

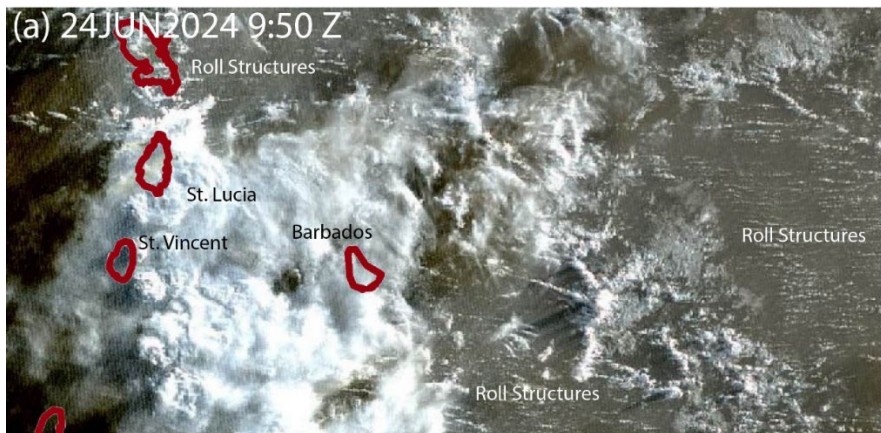

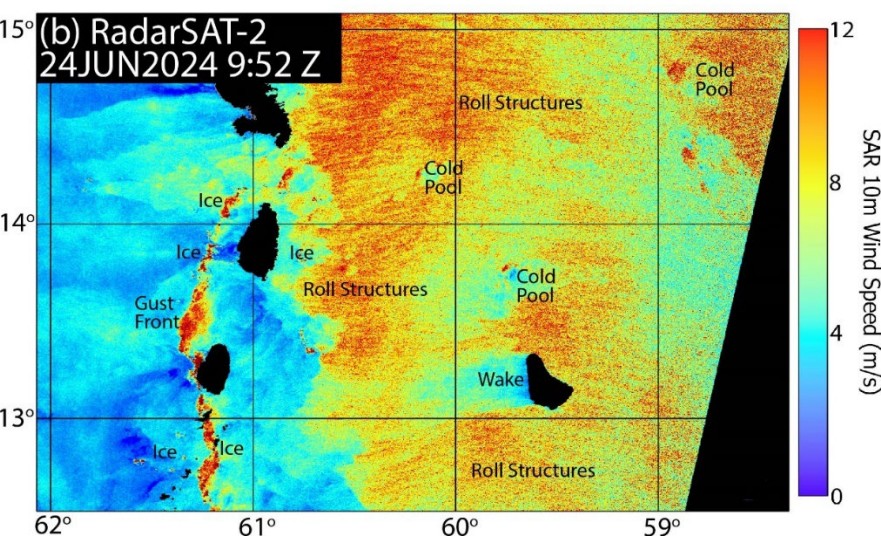


**Figure 1. June 24, 2024 is presented as an example day from the MAGPIE mission of typical subtropical Atlantic conditions around Barbados including the onset of a Saharan Air Layer event including an enhancement of an easterly jet. (a) Visible image from GOES-16 ABI at 9:50Z. Note the formation of cloud streets associated with the enhanced surface winds. (b) Corresponding RSAT2 Synthetic Aperture Radar (SAR) derived 10 m wind fields taken**

**at 9:52Z (Courtesy of Chris Jackson, NOAA STAR).**



The work presented here is a natural extension of *Park et al.,* (2022) who noted the significant amount of sampling time required for a "perfect measurement" dataset convergence of measured fluxes compared to a known source strength. In this work, we examine Lagrangian particle representation in a simple MABL like a cumulus field on the eastern side of Fig. 1 to demonstrate aspects of individual particle lifecycle to give insight into how to interpret boundary layer observations. The trajectory and residence time of a 10 μm aerodynamic diameter aerosol particle is evaluated for multiple stabilities based on surface heat flux and geostrophic wind speed, corresponding to a normalized Monin-Obukhov length scale. 10 μm was chosen as its deposition rate is noticeable at 3 mm/s or 11 m/hr, but not a dominant factor in particle trajectories relative to, say, mixing velocities on the order of 10s of cm/s to ~m/s.   Thus, these simulations can be informative for fine and larger coarse mode particles alike.  While Lagrangian studies have been made before (e.g., Shpund et al., 2011), these have been done in a 2-D framework. We analyze the particle concentration distribution for several stabilities and wind speeds associated with changes in the normalized Monin-Obukhov length scale. Lastly, given that the community and literature concerning particle diffusion is largely based on Eulerian models, we also examine model responses in 1D Eulerian simulations as a proxy to how most transport models may differ. We conclude with a discussion of how these findings can be applied to close measurements and models.

## 2 Model formulation and definitions

The primary model used here is the National Center for Atmospheric Research (NCAR) Turbulence with Lagrangian Particles (NTLP) model (*Moeng* 1984; S*ullivan and Patton* 2011; *Park et al.* 2020). The core ensemble of simulations uses a setup similar to Park et al. (2022) in that it employs a relatively small domain with periodic boundary conditions to consistently evaluate particle lifecycle over a range of wind speeds and stabilities.  However, this paper primarily focuses on tracking the particle's spatial evolution from their creation through the MABL and subsequent removal. Finally, we compare our findings with a 1D column-based transport model (the Marine Boundary Layer Aerosol Model, MARBLES; Caffrey, Hoppel, and Shi 2006), that uses turbulent mixing coefficients to predict vertical concentration profiles based on the NTLP simulations.

### 2.1 Large eddy simulations

The Eulerian LES component of NTLP solves the filtered velocity and potential temperature fields that drive the Lagrangian particle transport. Eulerian features from the LES are also supplied offline to the 1-D MARBLES model (Section 2.5) for comparison of particle statistics. At its core, the Eulerian LES is governed by mass, momentum, and energy conservation under the Boussinesq approximation:

$$\frac{\partial \tilde{u}_i}{\partial x_i} = 0,$$



$$\frac{\partial \tilde{u}_i}{\partial t} = -\frac{\partial \tilde{u}_i \tilde{u}_j}{\partial x_j} - \frac{\partial \tau_{ij}}{\partial x_j} + \frac{g\delta_{i3}}{T_0}\tilde{\theta} - \frac{1}{\rho_0}\frac{\partial \tilde{p}}{\partial x_i} + f(U_g - \tilde{u}_1)\delta_{i2}, \tag{1}$$

$$\frac{\partial \tilde{\theta}}{\partial t} = -\tilde{u}_i \frac{\partial \tilde{\theta}}{\partial x_i} - \frac{\partial \tau_{\theta i}}{\partial x_i},$$

where $f$ is the Coriolis parameter; $\tilde{u}_i$ is the resolved velocity vector; $\tilde{\theta}$ is the resolved potential temperature; $\tau_{ij}$ is the subgrid stress tensor; and $\tau_{\theta i}$ is the subgrid flux of potential temperature. A Poisson equation is solved to provide a divergence-free velocity field that is enforced by solving for resolved pressure $\tilde{p}$. The subgrid-scale turbulent fluxes are parameterized with a prognostic equation that solves for the subgrid-scale turbulent kinetic energy, which is then used to define a mixing length (Deardorff 1980). A constant, uni-directional geostrophic wind ($U_g$) is imposed to force the model via a pressure gradient. The model field is doubly periodic with a uniform grid in the horizontal ($x$ and $y$) directions; we employ a non-uniform grid in the $z$-direction to better resolve the upper mixed/mixing and surface layers. A pseudo-spectral discretization is used for spatial gradients in the horizontal directions, and a second-order finite difference is used for the vertical direction. Time integration is done using a third-order Runge-Kutta method, and a divergence-free filtered velocity field is maintained through a fractional step method. The lower boundary condition is set through a rough-wall Monin-Obukhov similarity relation, using a constant aerodynamic roughness length representative of the open ocean ($z_0 = 0.001$m).

**2.2 Lagrangian particle method**

Along with the LES of the Eulerian velocity and potential temperature fields, NTLP solves the trajectory of Lagrangian particles as they are transported throughout the MABL — a natural framework for understanding aerosol dispersion. The particle sizes are set to 10μm in aerodynamic diameter to represent coarse mode particles, which is much smaller than the smallest turbulence scales of the flow, allowing effects of particle inertia and other forces to be neglected (Balachandar and Eaton 2010). The size should be informative for our scale analysis from fine to larger coarse mode particles alike (e.g., results for transport will generally hold for the fine mode, except with less dry scavenging). The resulting equations governing the particle motion are:

$$x_{p,i}(t + \Delta t) = \mathbf{x}_{p,i}(t) + \mathbf{v}_{p,i}\Delta t + \eta\sqrt{2\overline{K}(x_{p,i})\Delta t} + \frac{\partial \overline{K}(x_{p,i})}{\partial z}\Delta t, \tag{2}$$

$$v_{p,i} = \tilde{u}_{f,i} - w_s\delta_{i3} \tag{3}$$

where $x_{p,i}$ is the position vector of particle $p$, $v_{p,i}$ is its velocity vector, $w_s$ is the Stokes terminal velocity of the particle, and $\Delta t$ is the timestep of the simulation. The last two terms in the equation of particle p's location account for unresolved motions in the LES: $\eta$ is an independent and identically distributed random value from a normal distribution, and $\overline{K}(\mathbf{x}_p)$ is



the horizontally averaged subgrid momentum diffusivity obtained from the LES model. Trilinear interpolation from the surrounding grid points is used to obtain Eulerian data at the particle location.

Here our focus is on transport out and back into the lowest model levels which are below 10m; a full consideration of surface layer transport, including the effects of waves and near-surface turbulence, remains a challenge for future numerical and observational studies. The effects of waves at the lower boundary are assumed to be captured entirely by a single constant roughness length. This is known to be a crude approximation (Edson et al. 2013) and neglects the effects of waves on very-near-surface particle transport (Richter et al. 2019); however, for the purposes of analyzing Lagrangian statistics of particles

throughout the region above the surface layer, these impacts will be marginal. Additionally, we ignore aerosol condensation/evaporation/swell (Winkler 1988) so that we can focus entirely on the Lagrangian transport, and since we simulate a dry MABL, we also ignore two-way coupling of momentum and energy between the aerosol particles and the air, which only become significant at high mass loadings, such as in clouds or in the spray-laden surface layer at high winds (Mellado 2017; Peng and Richter 2019).

For any timestep in the simulations, if a particle crosses the bottom boundary of the domain ($x_{p,3} < 0$), the particle is removed from the simulation, representing dry deposition onto the water surface. The dry deposition velocity is equal to the terminal velocity $w_s$. Our particle size is assumed to be 10 μm in aerodynamic diameter, towards the upper size of what is considered the coarse mode. For reference, such particles have a settling velocity of 0.3 cm/s or 11 m/hr. This oversimplification of dry deposition is primarily dependent on the Monin-Obukhov similarity theory and ignores the many

unaccounted physical processes with respect to surface deposition (impaction, e.g., Wang et al. 2017; coalescence, and local wave characteristics). The current framework is meant to simply interpret particle transport throughout the marine boundary layer, subject to an idealized treatment of their deposition. We expect there to be significant sensitivity to model resolution and configuration near the lower ocean surface. But such sensitivity will not impact our general results for above ~ 20 m.

**2.3 General NTLP simulation configuration and key metrics**

The overall simulation configuration is similar to that of Park et al. (2022), with the primary differences being mentioned here. The baseline domain is fixed at $10 \times 5 \times 0.8$ km ($x \times y \times z$) with $140 \times 140 \times 140$ grid points and periodic boundary conditions in the horizontal directions, corresponding to along wind (streamwise) $\Delta x = 70$m and crosswind (spanwise) $\Delta y = 35$m; we employ a stretched, non-uniform grid in the vertical $z$-direction to better resolve the surface and entrainment layers. For reference, the three grid centers at the bottom of the domain are roughly 1.5 meters apart at 0.63, 1.9, and 3.4

meters; the central boundary layer grid centers are roughly 27 meters apart at 286.5, 313.5, and 339.5 meters; finally, the imposed inversion centers are roughly 4 meters apart at 568.4, 572.3, and 575.8 meters. This configuration is consistent with previous convective atmospheric boundary layer studies (Moeng and Sullivan 1994; Sullivan and Patton 2011; Salesky et al. 2017; Park et al. 2022).



We consider three forcing regimes of the MABL: neutral, slightly unstable, and unstable - typical of most maritime environments. These regimes are characterized by shear-driven turbulence (neutral), followed by an intermediate balance of shear and buoyancy-driven turbulence that are the impetus of roll structures for both slightly and unstable stratification (Salesky et al. 2017). Stable conditions are excluded from this particular analysis as they require specialized model configuration and have a very high sensitivity to initial conditions.

As a reference for the temporal scales presented in this study, we define a characteristic eddy time scale as $T_{eddy} = z_{inv}/w_*$, where: $z_{inv}$ is the inversion height calculated as the maximum of the planar average of the potential temperature gradient (*Sullivan et al.,* 1998). The convective velocity scale $w_* = (\frac{g}{T_0}Q_0 z_{inv})^{1/3}$ is a function of surface heat flux $Q_*$, $z_{inv}$, gravitational acceleration $g$, and the reference temperature $T_0$ (Deardorff 1972). In this formulation $T_{eddy}$ is a characteristic timescale for a parcel of air to go from the surface to the top of the MABL as transported by a large convective eddy. For the neutral stability, we require a different characteristic timescale, defined as $T_{neut} = z_{inv}/u_*$, where $u_*$ is the friction velocity. To describe the relative roles of buoyancy and shear production of turbulence in the context of the convective boundary layer, we use the dimensionless Obukhov stability parameter $-z_{inv}/L$, where $L = \frac{-u_*^3 T_0}{\kappa g Q_0}$ is the Obukhov length scale. Here, higher positive values indicate stronger unstable conditions, with 0 indicating neutral conditions. The von Kármán constant $\kappa$ equals 0.4. Please note that negative values of $L$ indicate unstable conditions, zero indicate neutral, and positive values indicate stable conditions. At the beginning of the simulation, the velocity is set to $U_g$ throughout the domain, while the potential temperature is uniformly set to 300K under the inversion. Above $z_{inv}$, the potential temperature gradient is set to 0.05 K/m to restrict any significant growth of the boundary layer height, which remains located at approximately 570m throughout the simulation. At the upper computational boundary, a radiation condition is applied to prevent reflections back into the domain (Klemp and Durran*, 1983)*. The unstable configuration results in $T_{eddy} \approx 13$ minutes; that is, 13 minutes is a characteristic timescale for a parcel to reach the top of the MABL (at 9 m/s wind speed, this corresponds to ~ 7 km of horizontal advection during that time). All simulations were first run without particles for approximately 1 hour (e.g., corresponding to $4.7T_{eddy}$ for the unstable case) to allow the turbulence to reach quasi-stationary conditions. After this spin-up time, we initiate a puff emission of 30000 coarse-mode sea-spray aerosol particles randomly within the first grid box throughout the horizontal extent of the computational domain.

**2.4 Sets of simulations**

Two sets of simulations were performed. The first set, examined in depth in Section 3, consists of three baseline runs with a fixed geostrophic wind speed of 12 m/s and surface heat fluxes of 0, 2.5 and 24.6 W/m² (Table 1). These correspond to air-sea temperature differences of roughly 0, 0.15, and 1.3 ºC, respectively (using positive as more unstable convention). For the unstable conditions, we chose a heat flux equivalent to typical background air-sea temperature differences over sub-tropical to mid latitude oceans (*Quinn et al. 2021*). The difference between neutral and just slightly unstable is illustrative of how such a small perturbation in heat flux can make a significant difference in flow conditions; in this case a mere 0.15ºC. The





total duration of these simulations after spin up is 6000 seconds (100 min), or around 8 large eddy turnover times and ~ 60 km of advection for the unstable configuration. Particle statistics are saved every 5 seconds. In Section 4, we expand to an ensemble set of 25 runs to provide statistics across a wider range of conditions. Geostrophic wind speeds range from 4 to 20m/s with a surface heat flux from 0 to 36.9 W/m$^2$; this simulation set also includes the baseline heat flux and wind

configurations of Section 3 (Tables 2 and 3). Here simulations are 75000 seconds (about 20 hours) with particle statistics outputted every 100 seconds. This corresponds to ~700 km of advection at 12 m/s forcing.

**Table 1 Properties of numerical simulations used as a baseline in this study. Included are geostrophic velocity ($U_g$), 10-meter surface winds ($U_{10}$), surface heat flux ($Q_0$), marine boundary layer inversion height ($z_i$), Obukhov length ($L$), dimensionless Obukhov stability parameter ($-z_{inv}/L$), friction velocity ($u_*$), convective velocity scale ($w_*$), and**

**the characteristic eddy timescales ($T_{neut}$ for the Neutral simulation and $T_{eddy}$ for the Slightly Unstable and Unstable simulations).**

| Simulation | $U_g$ (m/s) | $U_{10}$ (m/s) | $Q_0$ (W/m$^2$) | $\Delta T$ (°C) | $z_{inv}$ (m) | $|L|$ (m) | $-z_{inv}/L$ | $u_*$ (m/s) | $w_*$ (m/s) | $T_{neut}/T_{eddy}$ (Sec) |
|---|---|---|---|---|---|---|---|---|---|---|
| Neutral | 12 | 9.2 | 0 | 0 | 570 | $\infty$ | 0 | 0.24 | - | 2313 |
| Slightly Unstable | 12 | 9.9 | 2.5 | 0.13 | 570 | 761.6 | 0.7 | 0.27 | 0.34 | 1633 |
| Unstable | 12 | 10.7 | 24.6 | 1.3 | 570 | 110.9 | 5.1 | 0.31 | 0.74 | 766 |

**Table 2. Same as Table 1 but for the ranges of ensemble set of simulations. A full table of simulations is provided in**
**Table 3.**

| $U_g$ (m/s) | $U_{10}$ (m/s) | $Q_0$ (W/m$^2$) | $\Delta T$ (K) | $z_{inv}$ (m) | $|L|$ (m) | $-z_{inv}/L$ | $u^*$ (m/s) | $w^*$ (m/s) | $T_{neut}/T_{eddy}$ (Sec) |
|---|---|---|---|---|---|---|---|---|---|
| 4 | 3.4-3.8 | 0-36.9 | 0-5.59 | 570 | 6-$\infty$ | 0-97 | 0.10-0.13 | 0.34-0.85 | 672-5761 |
| 8 | 6.4-7.4 | 0-36.9 | 0-2.88 | 570 | 30-$\infty$ | 0-19 | 0.17-0.23 | 0.34-0.85 | 672-5761 |
| 12 | 9.2-10.7 | 0-36.9 | 0-1.97 | 570 | 76-$\infty$ | 0-8 | 0.24-0.31 | 0.34-0.85 | 672-5761 |
| 20 | 14.8-16.3 | 0-36.9 | 0-1.29 | 570 | 286-$\infty$ | 0-2 | 0.43-0.48 | 0.34-0.85 | 672-5761 |

## 2.5 MARBLES

Lastly, we compare the Lagrangian particle results to a baseline model: the Naval Research Laboratory's Marine Boundary
Layer Aerosol Model (MARBLES). MARBLES is a column 1-D transport model that uses turbulent mixing coefficients to predict vertical concentration profiles (Caffrey, Hoppel, and Shi 2006; Lynch et al. 2016; Morcrette et al. 2009). We ran MARBLES with turbulent vertical mixing, temperature, and pressure as specified by data from the LES configuration; the



surface wind speed was set at a constant 10 m/s corresponding to the baseline geostrophic wind of 12 m/s. The processes of turbulent mixing, gravitational settling, surface deposition by turbulence and diffusion, coagulation, and sea-salt aerosol

surface emission using a hybrid Monahan/Smith formulation (Caffrey, Hoppel, and Shi, 2006) were used in the simulations. MARBLES can simulate surface-emitted sea-salt aerosol particles across 39 size bins ranging from 4.6 nm to 53.7 μm. For comparison with the NTLP simulations, we present results from the size bin ranging from 9.99 to 12.7 μm (the midpoint being 11.3 μm) in section 4.2.

## 3.0 Results

Our findings are best illustrated by first examining the three baseline simulations in depth: neutral, slightly unstable, and unstable (Table 1).  In this section we will systematically review the overall MABL turbulence structure, provide examples of single particle transport, and present overall particle lifecycle statistics.

*3.1 Turbulence characteristics.*

We begin with an overall description of the flow state and corresponding turbulence for the baseline cases as outlined in Table 1 and depicted in Figs. 2 and 3. The slightly unstable and unstable conditions result in Obukhov lengths of ~ 700 and 100 m, respectively (Stull 1988). The turbulence kinetic energy (TKE) is presented in Fig. 2(a) along with the vertical velocity variance $w'^2$. The TKE reaches a local maximum near 3m and decreases toward the central boundary layer (300m). Then it reaches another local maximum near 500m until it drops to zero at the inversion. The vertical velocity variance starts

at 0 at the surface and then increases to a maximum at the central boundary layer (300m), decaying back towards 0 as it approaches the inversion. It is evident that close to the surface, the TKE is dominated by variance in the horizontal wind velocity fluctuations. For all stabilities, the near-surface turbulence is characterized by long streaks, whose properties vary with height and stability (see Fig. 3).

In Fig. 2(b), the horizontally and temporally averaged wind speed $\sqrt{\overline{U}(z)^2 + \overline{V}(z)^2}$ (black line) shows the expected drop in

wind across the 570 m inversion. The averaging operator $\overline{(\ )}$ indicates temporal averaging from the start of the primary aerosol pulse to the end of the simulation. Under neutral conditions, a 12 m/s geostrophic flow corresponds to 8.2 m/s at 10m; this increases to 9.2 and 9.6 m/s for the weak and unstable cases, respectively.  There is a slight crossover in wind speeds at the top of the mixed layer, with neutral having a lower wind speed at 9.2 m/s, and unstable at 10.7 m/s.  The resolved shear- (S, Orange) and buoyancy-induced (B, Blue) turbulence production, defined as

$$S = - \left[ \overline{\widetilde{u'w'}} \frac{\partial U}{\partial z} + \overline{\widetilde{v'w'}} \frac{\partial V}{\partial z} \right], \tag{4}$$

$$B = \frac{g}{T_0} \overline{\widetilde{w'\theta'}} \tag{5}$$




also vary as expected with increasing instability (e.g., Moeng and Sullivan 1994). In the neutral case, turbulence is produced solely by shear production, while in the two unstable cases, buoyancy production exceeds shear production for increasing fractions the MABL interior. Specifically, buoyancy exceeds shear turbulence production between 195 and 422 m for the slightly unstable case, and between 51 and 454 m for the moderately unstable case (see Fig. 2(b)).

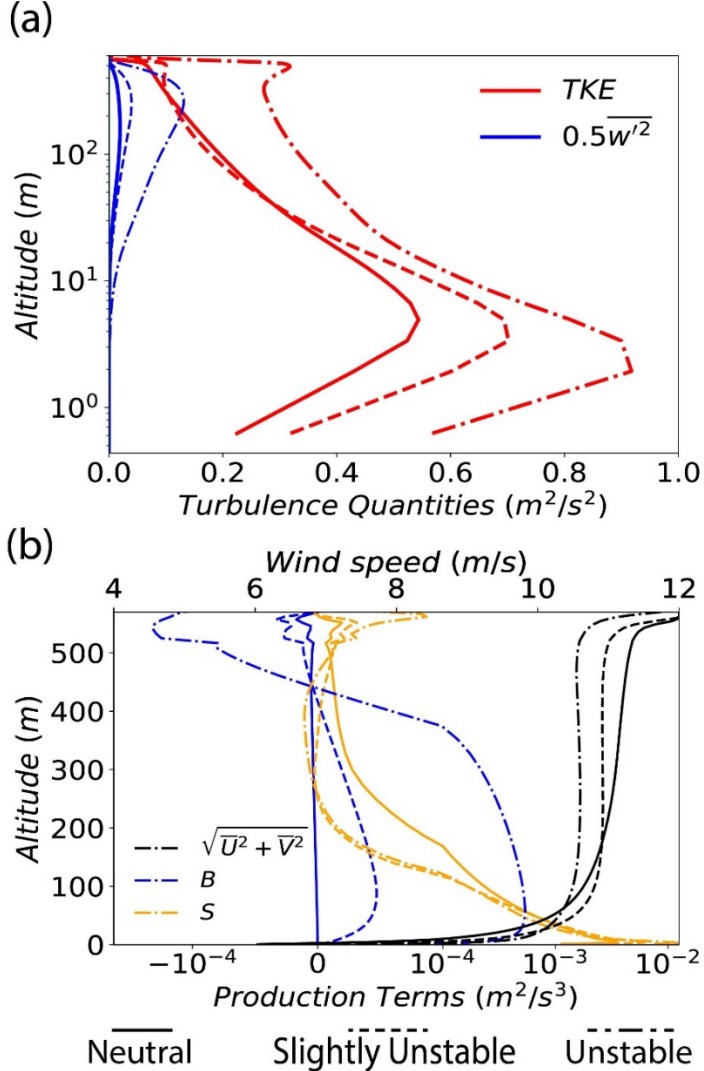

**Figure 2** NTLP model flow characteristics of (a) turbulence kinetic energy (TKE; red) along with the halved vertical velocity variance that is a component of the TKE (blue) and (b) wind speed $\sqrt{\overline{U}^2 + \overline{V}^2}$ (black), buoyant turbulence production B multiplied by 10 for visual comparison (blue), shear turbulence production S (orange) on a log-scale plot. The solid, dashed, and dash-dotted lines for both plots represent neutral, slightly unstable, and unstable, respectively.







*Figure 3.* **Instantaneous 10 m wind speed (or cup speed) and vertical velocity at 10, 80, 300, and 500 m for the baseline**
**neutral (left column), slightly unstable (center column) and unstable runs (right column).**



In Fig. 3, we show the 10 m horizontal instantaneous wind speed (or "cup speed") in the top row, and in the subsequent rows, we show instantaneous vertical velocity contours at 10, 30, 80, 300 and 500 m altitudes. For all three stabilities, as expected near the surface at $z = 10$m (first two rows of Fig. 3), the streaks of low cup speed correspond to higher values of
$w'$, indicative of ejections/upwelling of low-momentum fluid from below. Conversely, areas of enhanced cup speed correspond to downdrafts which bring high-momentum fluid downwards. These wind streaks, approximately 2 km apart peak to peak, match well in scale to the wind streaks observed in Fig. 1. At higher altitudes, the non-neutral cases provide starkly different structure (e.g., *Salesky et al. 2017*), and as shown later, different mechanisms of particle transport. At 30 m, the lowest altitude typically measured by light aircraft and coincidentally also the peak in TKE for any unstable condition
(e.g., Fig.2 (b)), the streaks in $w'$ are quite clear. At 80m (roughly the lowest for larger research aircraft) the full structure of coherent roll structures is sharply defined in narrow updrafts and broad downdrafts. These structures correlate with signature peaks near the center of the boundary layer, or $z = 300$m. These streaks that elongate in the spanwise direction are indicative of convective roll structures and have been documented in other studies (*Moeng and Sullivan 1994; Salesky et al. 2017*). Indeed, even the slightest addition of a heat flux in our slightly unstable case, is enough to amplify these distinct
features. In both unstable cases, these roll structures are centered approximately 1-2 km apart in the spanwise direction. Near the top of the mixed layer, the streaks weaken in favor of individual plumes but nevertheless maintain a coherent form of structure that can be visually traced back to the orientation of convective rolls.

The above inspection of the turbulent system illustrates the prototypical turbulence structure of the neutral and unstable atmospheric boundary layer, and we will now shift toward interpreting this system as a medium of primary aerosol particle
transport. We expect the levels of turbulence, and particularly the differences between the coherent structures of the neutral and unstable boundary layers to play a role in particle behavior. This is explored starting in Section 3.2.

*3.2 Examples of particle transport*

To visualize primary aerosol particle transport, we illustrate 100 particle trajectories for each of the baseline neutral, slightly unstable, and unstable cases in Figs. 4(a), (b), and (c), respectively. These trajectories were generated with output every 5
seconds for 6000 seconds (i.e., 100 minutes). This time corresponds to different spatial scales: at 12 m/s geostrophic wind (9.5 m/s within the mixed layer), this corresponds to ~60 km of horizontal transport distance. In terms of the eddy turnover time, 6000 seconds corresponds to $T_{neut}$ of 2.5 for the neutral case, and a slightly/moderately unstable $T_{eddy}$ of 3.7 and 7.8, respectively. Clearly visible are multiple particle vertical transport regimes as instability grows. For neutral and slightly unstable conditions, where vertical velocities are generally low relative to the unstable case (see Figs. 2 and 3), particles can
maintain any given altitude range for over an hour. These plots likewise demonstrate the behavior near the surface: for all stabilities, particles can linger near the surface for 30 minutes or more, and for the neutral and weakly unstable cases, many particles can reside there for hours without ever exiting the near-surface environment. In reality, we would expect wave action to further enhance mixing near the bottom boundary, but this is a process that cannot be captured by the current LES model. Nevertheless, these plots demonstrate the nominal speeds and distances of the mixing process.



For a particle to be transported to a height where aircraft could potentially sample it, it would first need to "escape" the near-surface layer noted above. In this sense, Fig. 4(a) and (b) show multiple examples of particles which fail to do this, and linger at low altitudes before ultimately depositing back on the surface. Other particles, however, quickly leave the near-surface region and get swept up in MABL-scale coherent updrafts. In all three cases, two primary modes of primary aerosol transport are visible: those that linger near the surface, and those that escape to be transported across the entire MABL.


**Figure 4. Individual particle trajectories for the three baseline simulations with 12 m/s geostrophic wind forcing: (a) Neutral, (b) Slightly unstable, and (c) Unstable stratification. Each panel shows selected particle paths (colored lines) and the full ensemble of trajectories (gray lines) over 6000 second period. The top x-axis represents the corresponding large eddy turnover times ($T_{neut}$ and $T_{eddy}$), and bottom x-axis shows time in seconds.**






For neutral stability, it is possible for particles to be transported vertically and then remain at a near constant altitude for a considerable amount of time (see Fig. 4(a)). This is generally true for the slightly unstable case as well (Fig. 4(b)), but for the unstable case, we see that particles can no longer linger in the near-surface region for long before being lofted upwards or

deposited back on the surface, and those particles that escape are seen to undergo more cyclic motions. This behavior is characteristic of the MABL-scale convective up- and downdrafts. The distribution of updrafts and downdrafts generally prevent particles from lingering for more than roughly 1000s (roughly a single $T_{eddy}$) at any one height, and thus their deposition back onto the ocean surface is regulated by how close they get as a result of the periodic downdrafts. By contrast, the neutral and slightly unstable cases see particles able to linger for much longer periods of time at one height, even within

the interior mixed layer of the MABL. This is a direct consequence of the varying turbulence structures seen in Fig. 3.

Altitude transitions for particles in the neutral and weakly unstable cases are typically slow, with the vertical motions associated with convective updrafts only participating with stronger instability. Nevertheless, statistically speaking there are some particles that can rise to the top of the MABL in 10 minutes, even in the absence of convective updrafts. For example, the light gray particle in Fig. 4(a) contradicts the norm by reaching 500m in 30 minutes, returning to 100m around 50

minutes later, and remaining there until the end of the simulation. For unstable conditions (e.g., Fig 4. (c)), these rapid ascents happen much more often, since particles can ride updrafts to quickly reach higher altitudes (consider the orange particle for example). For particles across all stabilities, descents sometimes bring the particles back to the near surface environment, where the particle can either deposit at the surface, linger for some time, or re-emerge upwards into the central MABL and take another excursion. In the unstable case, the frequency of proximity to the lower surface is higher due to

their cyclic nature.

Overall, these physical observations of Lagrangian trajectories through a heterogeneous turbulent system act as the foundation for a more quantitative statistical analysis expanded further in Section 3.3.

*3.3 Statistical descriptions of particle behavior*

Analysis of Fig. 4 reveals that while each particle has a unique trajectory in its lifecycle, there are distinct patterns or modes

of vertical transport that align with the flow fields in Fig. 3. These patterns are important for interpreting particle measurements made *in situ* by, say, aircraft that are sampling at a fixed height, since it suggests an individualized history that should be considered. Here, we aggregate these unique trajectories to better observe the overall statistical transition through altitude. Fig. 5 presents this analysis as time series of particle concentration with height, normalized by the initial number of particles (Fig. 5(a) neutral; (b) slightly unstable; (c) unstable).

For neutral conditions, as expected, we observe a plume slowly dispersing in the time-height plane, taking approximately 100 minutes (or ~50 km) to reach 500 m. Despite this extensive transport, the highest concentrations remain near the surface where particles ultimately deposit. The neutrally stratified MABL, lacking strong up- and downdrafts, limits rapid altitude transitions which can be seen in the low particle concentration throughout the MABL. Even with the slightest instability,




however, along with the onset of coherent roll features, changes qualitatively particle transport mode, as seen in Fig. 5(b).

Now, a small plume of particles detaches from the high concentration region near the surface, and ascends to the MABL top in roughly 30 minutes (or ~20 km). This plume is eventually mixed downwards, leading to a peak in concentration in the MABL center at later times. This pattern of vertical transport in a roll feature is reminiscent of the looping phenomenon to the inversion top followed by fumigation (e.g., Stull, 1988).

These features are amplified in the unstable case (Fig. 5(c)), where a much more prominent particle plume now hits the

MABL top in under 1200 seconds (20 min or ~10 km). This, combined with the rapid deposition of particles that remain at the surface, leads to a clear peak concentration near the MABL top which persists from $1500 < t < 2000$s, accompanied by a near-absence of particles near the surface. The concentration more quickly approaches a well-mixed state, owing to the increased strength of the convective MABL motions. Overall, these features in Fig. 5 indicate the potential for vastly different particle observations made downstream of an instantaneous event, over horizontal length scales of ~5 to 50 km.

While Figure 5 illustrates the particles' bulk transport in time, it also provides the foundation for understanding how their modes of transport dictate their deposition probability as the simulation progresses. There are two modes of particle transport identified above: a group of "unlucky" particles remains at the surface, while others are rapidly transported to the MABL top. In particular, we are concerned with the expected lifetime of these particles, as in practice this is essential information when trying to attribute measured particles to local vs. distant sources, or account for airborne chemical processing. We

anticipate that the distinct transport modes will imprint themselves on particle lifetime statistics, and that each particle's longevity is related to its path from the ocean surface.

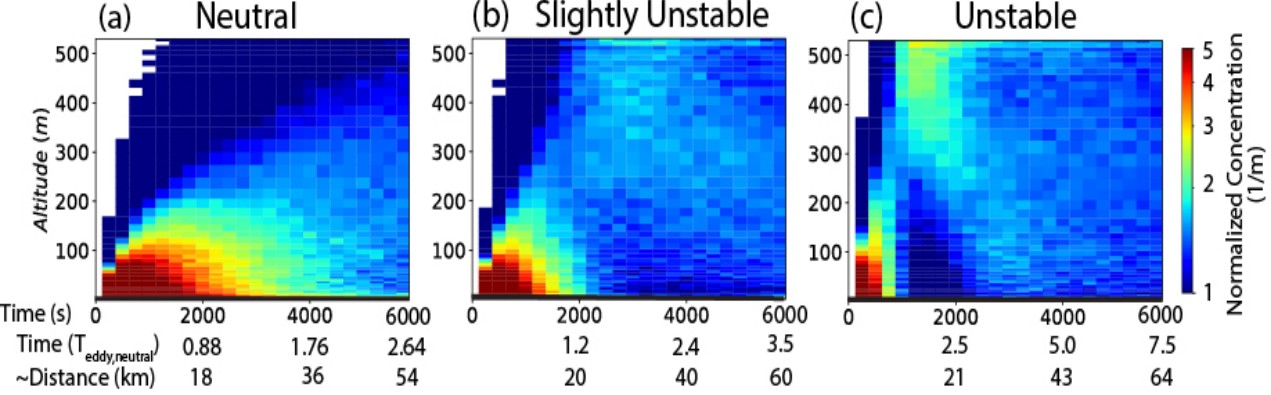

**Figure 5.** Time-height evolution of normalized particle concentration for the three baseline cases: (a) neutral, (b)
slightly unstable, and (c) unstable conditions, over 6000 seconds. The bottom axes show time in seconds, corresponding time in eddy turnover times and horizontal transport distance assuming 12 m/s of geostrophic wind speed.





To this end, Figure 6(a) plots the PDF of total particle residence time for the particles that deposited at some point during the

baseline simulations—note that panel (a) shows the baseline simulations ran for 1250 min (~20 hours). Over this time period, the simulations show survival rates of 6%, 4%, and 2% for the neutral, slightly unstable, and unstable atmospheric conditions, respectively, meaning that this fraction of the original 30000 particles were left at the end. Figure 6(a) indicates that the most probable life expectancy is less than 1000s, corresponding to the first transport mode where particles in this "puff" experiment remain near the surface before quickly depositing—consistent with Fig. 5. As conditions shift from

neutral to slightly unstable, a subtle local minimum emerges around 3000 seconds, which is indicative of the second transport mode. For the unstable case, the most likely lifetime is still less than 1000s, but the local minimum corresponds to the depleted near-surface region clearly seen in Fig. 5(c), while the plume is hitting the MABL top. At this stage, near surface particles have already deposited, while particles at the top must first descend before potential deposition. This renders a certain time range unlikely for particle lifetimes, ~2000s for the unstable case and 4000s for the slightly unstable

case. As the plume which reached the MABL top descends and is mixed throughout, a second peak in the particle lifetime PDF begins to emerge as these become available again for deposition.

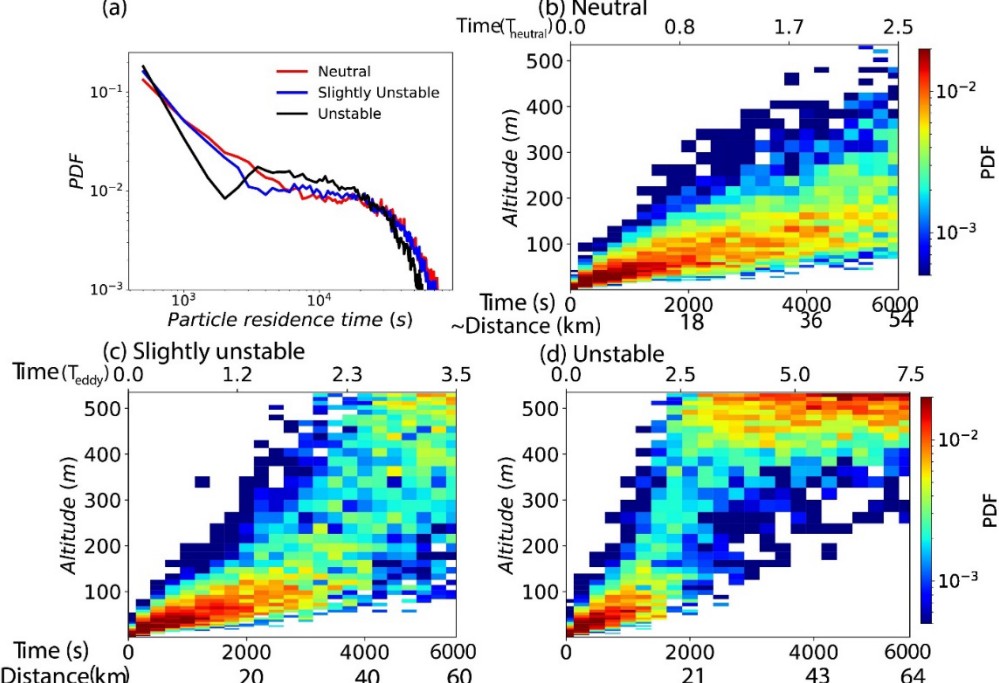

**Figure 6. (a) Probability density function (PDF) of particle residence times for the three baseline cases of neutral**

**(red), slightly unstable (blue), and unstable (black) conditions run for 1250 min (~20 hours). Panels (b)–(d) show the time-height evolution of the column count normalized deposited particle concentration for (b) neutral, (c) slightly unstable, and (d) unstable cases. The top x-axis of (b–d) shows the respective eddy turnover time ($T_{neut}$ and $T_{eddy}$), and bottom x-axis shows time in seconds horizontal transport distance assuming 12 m/s of geostrophic wind speed.**



The presence (or lack thereof) of the bimodal residence time distribution can also be seen by directly computing the probability of a deposited particle's maximum altitude reached, given a particular lifetime; this is shown in Fig. 6(b,c,d). As expected for any level of vertical transport, the longer a particle survives in the boundary layer, its maximum altitude reached during that lifetime increases. For example, in the neutral case, particles which lived to 6000 s (100 min) all reached higher than 80 meters, whereas particles which only lived 2000 s (~30 min) have a most likely maximum height of 80

meters, with a wide variability around this point. Indeed, the neutral case exhibits a smooth, monotonic relationship between maximum altitude attained and time of deposition due to its lack of strong vertical updraft structures, resulting in deposited particles not clearly distinguishing the first and second modes of particle ensemble transport. On the other hand, the unstable case demonstrates a sharp shift in maximum height probability at 2000 s (or 2.5 $T_{eddy}$), indicating that particles deposited at or beyond 2.5 $T_{eddy}$ entered convective rolls which rapidly transport near the top boundary layer and back down to the near-

surface. Some particles do not fully reach the top of the boundary layer during this process (i.e., there are appreciable probabilities beneath 500 m beyond 2000 s). Nevertheless, the second mode of particle transport is clearly visible as the instability increases. This process emerges faster given higher instabilities and gives rise to the bimodal distribution of particle lifetime seen in Fig. 6(a).

A complementary quantity that can help shed light on this difference between the neutral and unstable distributions of

lifetime is the total time spent in different height ranges. Fig. 7(a-c) shows the probability distribution function for total time spent in each of the 100 m-deep height ranges starting from 10 m. Each time a particle enters a range, a clock is started until it leaves (either upwards or downwards), and this time interval is stored. The PDFs shown in Fig. 7 are the distribution of the lengths of these times spent. Several interesting features can be seen. Focusing first on the neutral stability, there is similarity between the PDFs at various heights, particularly among the interior height ranges (blue, cyan, orange). There is a local

minimum around 100 s, and a maximum lifetime of around 4000 s. The highest and lowest ranges (red and black, respectively) show an increased tendency of particles to linger longer, owing to the weakened vertical motions in these regions due to the proximity to the surface and the inversion layer. As convective instability is increased, the distinction between the interior and highest/lowest ranges becomes starker: the blue, cyan, and orange lines nearly coincide with one another, highlighting the fact that central mixed layer is indeed well-mixed; i.e., a particle has an equal likelihood of

spending time in each of these ranges. Consistent with Figs. 4, 5, and 6 above, even weak instability provides a mechanism by which particles can traverse the entire MABL, rendering their residence in the central portions nearly identical. With the strongest instability, the highest height range, which contains the inversion layer, yields the longest residence times, owing to the convection rapidly bringing particles to the top, which then linger for potentially long periods of time before descending. The time a particle spends in the boundary layer particle lingering is seen near the surface for the neutral and unstable

instabilities, but for the unstable case the convective rolls bring the particles too near the lower surface to last.





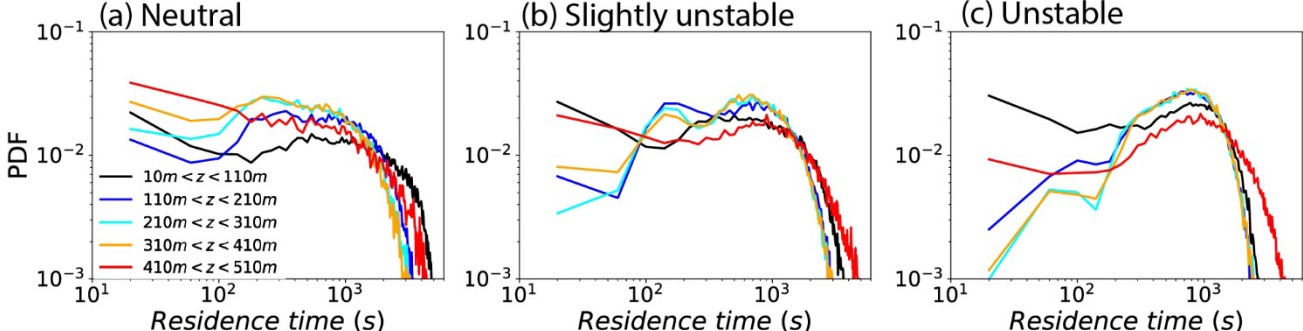

**Figure 7. Probability density function (PDF) of particle residence times within 100-m deep altitude layers for the three baseline cases of neutral (a), slightly unstable (b), and unstable (c). Different colors represent altitude layers from near-surface (10–100 m) to upper MABL (410–510 m).**

What emerges from this brief statistical analysis is a conceptual picture of primary aerosol transport that will be developed further below: in even weakly unstable conditions, coherent turbulent structures provide an efficient means for particles to reach the MABL top as they cycle up and down. Overall, the expected lifetime of particles in neutral and unstable boundary layers are similar, but for different reasons: neutral stratification yields particles that meander randomly and often do not reach the MABL top, whereas unstable stratification provides a mechanism for a particle to rapidly and repeatedly sample the entire MABL. This cycling process provides the basis for the modeling framework described below.

*3.4 Extended range of stability regimes*

As we have seen in Section 3.1, MABL turbulence, depending on stability, may exhibit convective rolls which lead to nonlocal transport of particles. The lack of coherent structures in a neutrally stratified boundary layer, by contrast, leads to qualitative changes in the shape of the particle lifetime distribution (even if the average lifetime is similar) and overall trajectory as observed in Section 3.2. The existence of (or lack thereof) these coherent structures are dependent on the combination of wind shear and surface heat fluxes (*Grossman* 1982; *LeMone* 1973; *Moeng and Sullivan* 1994). For example, the LES study of Salesky et al. 2017 systematically investigated the transition between roll- and cellular-type convection, spanning a wide range of the normalized Monin-Obukhov length scale $-z_{inv}/L$. Knowing that coherent structures provide starkly different pathways of particle transport, we briefly expand upon their effects based on the altitude-based PDFs as we did in Fig. 5, but for a more varied range of atmospheric forcing conditions.

An overview of these simulations is presented in Fig. 8, where the same quantity as presented in Fig. 5 is plotted – emission-normalized particle concentration. The number labeled on each plot represent the case numbers listed in Table 3. In Fig. 8, each descending row represents an increase in surface heat flux (0, 2.5, 6.2, 13.3, 24.6, 36.9 Wm$^{-2}$) whereas each column from left to right represents an increase of geostrophic wind speed (4, 8, 12, 20 m/s). From Sec. 3.3, and Fig. 5 in particular, we anticipate that the presence of strong convection will yield the two particle transport pathways described above: those




which linger near the surface and deposit, versus those which are quickly taken to the MABL top. These different routes lead to the bimodal particle lifetime distribution in Fig. 6, and can be seen in Fig. 5 as a peak in the concentration near the MABL top, coupled with a particle deficit near the surface at the same times.

**Figure 8. Normalized particle concentration for the 24 ensemble simulations with varying geostrophic wind speed (columns: 4, 8, 12, and 20 m/s) and surface heat flux (rows: 0, 2.5, 6.2, 13.8, 24.6, and 36.9 W/m²). Each panel shows the time-height evolution over 75000 seconds, with panel numbers corresponding to cases in Table 3.**



Generally speaking, Fig. 8 confirms this behavior, and shows clearly how this bifurcated particle transport emerges under certain combinations of instability and wind speed. Using $-z_{inv}/L$ as an indicator of the strength of shear- versus buoyancy-driven turbulence, we see that small values (upper right of Fig. 8) resemble the neutrally stratified behavior, while strong convection (lower left of Fig. 8) sees rapid ascent and a subsequent near-surface concentration void for a prolonged period of

495  time. For $U_g = 20$ ms$^{-1}$ (rightmost column), for example, this bifurcation emerges gradually, and the non-monotonic concentration evolution in time in the near-surface only emerges for Case 16 and beyond. For smaller values of $U_g$, the emergence of this near-surface void happens more rapidly, and a larger fraction of the particles making it to the MABL increases. Thus, we see clearly from Fig. 8 that particle transport pattern is heavily dependent on the strength of instability, and that even relatively weak unstable conditions can both rapidly bring particles to the MABL top but also mix them more

500  efficiently throughout the MABL. Recalling Fig. 6, this latter point emphasizes that the MABL stability does not necessarily change the mean particle residence time; rather, it greatly influences the spatial locations of sampling certain particles.

**Table 3. Full listing of experiments in Table 2. The boldfaced row of Case numbers represents the baseline simulations primarily discussed in Section 3.**

| Case number | $U_g$ | $U_{10}$ | $Q_0$ | $\Delta T$ | $|L|$ | $-z_{inv}/L$ | $u_*$ | $w_*$ | $T_{\text{neut}}|T_{\text{eddy}}$ |
|---|---|---|---|---|---|---|---|---|---|
| | m s$^{-1}$ | m s$^{-1}$ | W $m^{-2}$ | K | m | – | m s$^{-1}$ | m s$^{-1}$ | s |
| 1 | 4 | 3.4 | – | – | Inf | 0 | 0.10 | – | 5761 |
| 2 | 8 | 6.4 | – | – | Inf | 0 | 0.17 | – | 3251 |
| **3** | **12** | **9.2** | **–** | **–** | **Inf** | **0** | **0.24** | **–** | **2313** |
| 4 | 20 | 14.8 | – | – | Inf | 0 | 0.43 | – | 1308 |
| 5 | 4 | 3.7 | 2.5 | 0.36 | 59.3 | 9.4 | 0.12 | 0.34 | 1633 |
| 6 | 8 | 7.0 | 2.5 | 0.19 | 307.4 | 1.8 | 0.2 | 0.34 | 1633 |
| **7** | **12** | **9.9** | **2.5** | **0.13** | **761.6** | **0.7** | **0.27** | **0.34** | **1633** |
| 8 | 20 | 15.3 | 2.5 | 0.09 | 3070.9 | 0.2 | 0.43 | 0.34 | 1633 |
| 9 | 4 | 3.7 | 6.2 | 0.92 | 27.3 | 20.6 | 0.12 | 0.47 | 1209 |
| 10 | 8 | 7.2 | 6.2 | 0.48 | 140.1 | 4.0 | 0.21 | 0.47 | 1209 |
| 11 | 12 | 10.3 | 6.2 | 0.34 | 359.8 | 1.6 | 0.29 | 0.47 | 1209 |
| 12 | 20 | 15.8 | 6.2 | 0.22 | 1410.6 | 0.4 | 0.45 | 0.47 | 1209 |
| 13 | 4 | 3.8 | 13.3 | 2.84 | 15.0 | 37.6 | 0.13 | 0.59 | 960 |
| 14 | 8 | 7.2 | 13.3 | 0.97 | 75.6 | 7.5 | 0.21 | 0.59 | 960 |
| 15 | 12 | 10.4 | 13.3 | 0.67 | 193.2 | 2.9 | 0.29 | 0.59 | 960 |
| 16 | 20 | 16.1 | 13.3 | 0.43 | 684.8 | 0.8 | 0.45 | 0.59 | 960 |
| 17 | 4 | 3.8 | 24.6 | 5.01 | 8.5 | 67 | 0.13 | 0.74 | 766 |
| 18 | 8 | 7.3 | 24.6 | 1.92 | 43.1 | 13.2 | 0.22 | 0.74 | 766 |
| **19** | **12** | **10.7** | **24.6** | **1.32** | **110.9** | **5.1** | **0.31** | **0.74** | **766** |
| 20 | 20 | 16.6 | 24.6 | 0.84 | 391.5 | 1.4 | 0.47 | 0.74 | 766 |





| 21 | 4 | 3.8 | 36.9 | 5.59 | 5.9 | 96.5 | 0.13 | 0.85 | 672 |
| 22 | 8 | 7.4 | 36.9 | 2.88 | 29.9 | 19.2 | 0.23 | 0.85 | 672 |
| 23 | 12 | 10.7 | 36.9 | 1.97 | 75.5 | 7.6 | 0.31 | 0.85 | 672 |
| 24 | 20 | 16.3 | 36.9 | 1.29 | 286.2 | 2.0 | 0.48 | 0.85 | 672 |

## 4.0 Comparison to a 1D column-model (MARBLES)

To conclude this investigation, we compare particle dispersion in the MABL between our full 3D Lagrangian framework and a 1D column-based Lagrangian framework using an eddy-diffusivity-based parameterization. In most large-scale models, such as numerical weather prediction and climate models, aerosols transport and mixing of are typically represented by subgrid parameterizations based on eddy-diffusivity mixing coefficients. This simplified approach contrasts with our 3D Lagrangian framework, which fully resolves individual updrafts and downdraft features. This comparison demonstrates the fundamental limitations of representing three-dimensional turbulent transport processes through 1D eddy-diffusivity-based parameterizations—a challenge that impairs our interpretation of aerosol measurements in the context of meso to large-scale models verification or inversions.

As mentioned in Section 2.5, we use MARBLES, a 1D Lagrangian-based aerosol model (*Caffrey et al. 2006*), as a benchmark to evaluate how well mesoscale model parameterizations can represent aerosol transport in the MABL. Using the 1D MARBLES model that predicts concentration allows us to perform a consistent comparison of our Lagrangian framework findings with an established and well-characterized sea salt model. By design, a 1D model such as MARBLES cannot represent the 3D motions demonstrated in Fig. 3, but its formulation is meant to capture the horizontally-averaged behavior by accounting for subgrid turbulent dispersion by requiring turbulent mixing coefficients (or eddy diffusivity) $K_C(z)$. Therefore, to enable such a comparison, $K_C(z)$ is parameterized by the expression proposed by Nissanka et al. (2018) shown below in Eq. (3), which takes canonical Monin-Obukhov similarity theory to represent eddy diffusivity at the surface layer and extends it throughout the MABL. The formulation of Nissanka et al. (2018) blends Monin-Obukhov similarity theory near the surface with a standard approach for the mixed layer, and it also considers atmospheric stability (*Freire et al., 2016; Kaimal and Finnigan 1994; Nissanka et al. 2018*):

$$K_C(z) = \begin{cases} \dfrac{\kappa u_* z}{\phi(\zeta) Sc_t}, & \text{if } z < z_b, \\ a \dfrac{\kappa u_* z}{\phi(\zeta) Sc_t} \left(1 - \dfrac{z}{z_{inv}}\right)^2, & \text{if } z \geq z_b, \end{cases} \tag{6}$$

where $z_b$ is the surface layer height (taken as $0.1\, z_{inv}$) and $a = 1/(1 - z_b/z_{inv})^2$, that creates a continuous transition between the two parts of Eqn. (3). Here $\zeta = z/L$ is the stability parameter, $L = -u_*^3 \overline{\theta}_s / \left(\kappa g \overline{w'\theta'}_s\right)$ is the Obukhov length at the surface, and $\phi(\zeta)$ are the stability functions. Finally, $\kappa \approx 0.4$ is the von Kármán constant and $Sc_t = K_m/K_c$ is the turbulent Schmidt number where in previous works (e.g., *Freire et al. 2016, Nissanka et al. 2018*) a value of $Sc_t = 1.3$ was





adopted to account for numerical differences in the diffusivity of particles and momentum in the simulation. The friction velocity, and inversion height are obtained from the LES output data and used to calculate the eddy diffusivity as an input to the MARBLES model, which outputs concentration profiles.

Figure 9 compares the concentration profiles of MARBLES with NTLP at multiple times for the three stability conditions
discussed in Section 3: (a) neutral; (b) slightly unstable; and (c) unstable—all with a geostrophic wind speed of 12 ms[-1]. We focus particularly on times that were identified in Fig. 5 to be associated with the transition to particle cycling through the MABL. The rate of vertical mixing is clearly greater for NTLP than MARBLES, as evidenced by the higher concentration levels above the first grid point at 800s across all stability conditions. In both unstable cases, NTLP profiles show a concentration maximum at the MABL top—although at different times—consistent with the discontinuity seen in Fig. 5. By
contrast, MARBLES exhibits slower vertical dispersion and a subsequent rapid loss of aerosol particles after emission for all stability conditions. Rather than showing a local maximum of vertical particle concentration near the MABL top, MARBLES slowly diffuses particles towards a more well mixed state, even in the unstable case. Consequently, MARBLES overestimates the relative particle concentrations at the surface throughout much of the simulation.

The observed disparity between our Lagrangian LES simulation and MARBLES forced by the same LES model reinforces
the well-known conclusion that K-theory inadequately represents nonlocal mixing (like convective rolls) in unstable stratification via a gradient diffusion hypothesis (*Wyngaard 2010; Nissanka et al. 2018*). The Lagrangian trajectories show strongly non-local transport due to the convective motions of the unstable MABL, and the absence of numerical diffusion in the Lagrangian framework shows the tendency of column-based models to artificially smooth particle transitions and the boundary layer inversions. While it is not practical to model aerosol transport from a Lagrangian point of view in weather
and climate models, this approach reveals both the limitations of traditional approaches and provides insights for interpreting observations and time histories.

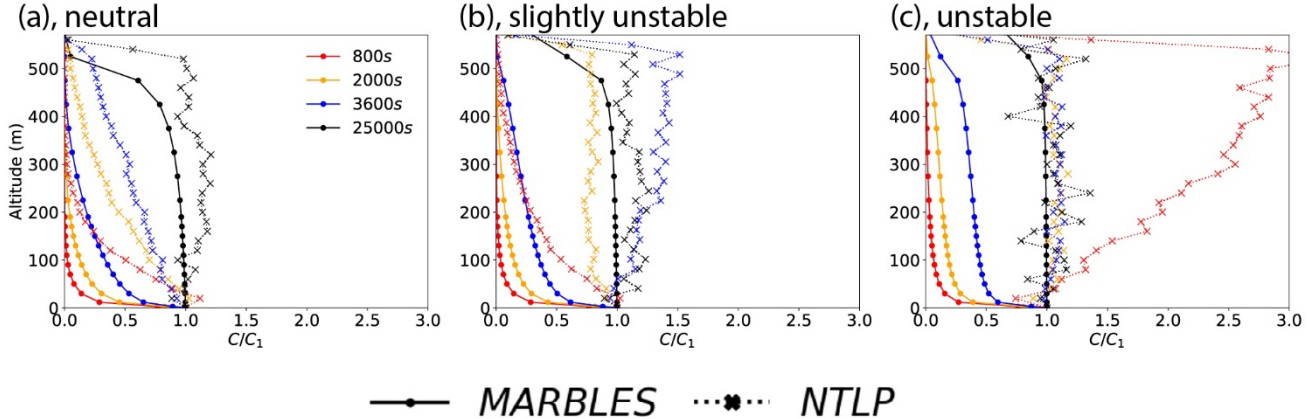

**Figure 9. Comparison of vertical concentration profiles between Lagrangian NTLP (dotted lines) and 1-D MARBLES (solid lines) for the benchmark cases (a) neutral, (b) slightly unstable, and (c) unstable cases using the turbulent**



**mixing coefficients from the parameterization proposed by Nissanka et al. 2018 (Equation 11) obtained using NTLP input variables. Profiles are normalized by $C_1$, the concentration at the first grid point.**

From a broader perspective, mesoscale models can capture some of the large-scale features shown in Figure 1, making them valuable for interpreting in-situ and airborne observations affected by aliasing problems. However, mesoscale models face

two key limitations: (i) they cannot resolve the detailed turbulent structures (Figure 3) that LES models can capture, and (ii) their aerosol transport is often unrealistic due to oversimplified parameterizations, as illustrated in Figure 9. These limitations hinder accurate projection of various observations into models. While significant efforts have been made to improve turbulence and cloud parameterizations in large-scale models (e.g., eddy-diffusivity mass-flux (EDMF) and PDF-based schemes) to better represent non-local transport in the MABL, these improvements must be extended to aerosol

parameterizations—potentially by coupling aerosol transport with these enhanced turbulence and cloud schemes. The community is starting to move in this direction (e.g., Li et al., 2021). Additionally, LES with mesoscale-size domains (i.e., ~150 x 150 km$^2$) and realistic configurations (e.g., Schulz and Stevens, 2023) can help identify which processes and features are compromised in mesoscale models due to poor representation of aerosol distribution and concentration, such as the rapid vertical transport of particles in convective updrafts, and consequent formation of concentration maxima near the MABL top,

detrained aerosol layer above the inversion layer, marine cloud morphology (e.g., roll- vs cellular-type structures), and precipitation development through aerosol-cloud interactions.

**5.0 Discussion and Conclusions**

Missions such as CAMP$^2$Ex, PISTON and MAGPIE demonstrated wind inhomogeneity that posed numerous questions on what specifically an aircraft, ship or towers measure relative to airmass history. To be sure, the MABL is a complex

environment; even in the "fair weather trade winds" frequent transients of easterly waves, Saharan Air Layers, gust fronts, channel flows, etc. results in significant wind variability over short temporal and spatial scales. These features led to numerous unknowns on a) What the spatial and temporal scale of mixing is for a specific event, for example a wind jet forming around an island or a ship emission; and b) What are the implications of dispersion on how it can be observed and be used to constrain model physics and through it tune parametrizations and infer fluxes. To this end, this study used a high-

resolution LES-Lagrangian framework to investigate near field primary aerosol particle transport trajectories released from the ocean surface. While this study is highly idealized it nevertheless provides insight into how a surface emission event realistically disperses and potential implications as to how particles can be observed.

We first simulated multiple stabilities given a single geostrophic wind speed that represented neutral, slightly unstable, and unstable boundary layer states. After qualitatively describing these states, we introduce primary aerosol particles in a

Lagrangian framework and observe their trajectories in which modes were discovered. These modes were then represented through statistical descriptions that help interpret the manner of their turbulent transport. It was found that the first mode of



particles represents ones that are relatively short lived and do not exit the near surface environment; they simply deposit back to the ocean. The second mode describe particles that enter the interior MABL and then circulate up and down along coherent MABL structured updrafts and downdrafts with more instability. Since atmospheric conditions can widely vary, we provided additional ensemble simulations that capture a range of stratified boundary layer flows, primarily based on changing surface heat flux and geostrophic wind speed. What these show is that for unstable conditions, an "empty zone" appears downstream of a source as those particles are transported upwards (so called "looping"), followed by more downward diffusion in spatiall larger but weaker downdrafts (so called "fumigation"). The results suggest that surface heat flux and geostrophic wind speed contribute to the manner of vertical transport as well as the deposition rate of aerosol particles. The level of vertical transport is dictated by the interplay between shear and buoyancy-driven turbulence that affect the residence lifetime and distribution of aerosol particles.

We concluded our investigation by comparing the current Lagrangian model with the 1D, column-based MARBLES model forced by our LES simulation to expand upon the contrasting results of predicting vertical concentration profiles. In this situation MARBLES behaves like a subgrid model for a mesoscale simulation. This comparison showed vastly different near-field time evolution of not only particle dispersion, but also particle lifetime owing to subgrid scale parametrizations in the lowest levels during release. While this comparison can be considered "fair" we nevertheless recognize that what these simulations show is the extreme sensitivity of model outcomes to particle transport algorithms and meteorology. Just as has long been adapted for cloud physics, aerosol models should probably initially account for mass fluxes versus simple eddy diffusivity.

Indeed, an outstanding challenge to this study, and nearly all micro to mesoscale meteorology simulations, are the postulated sub-grid parametrizations, particularly near the ocean surface. As particles are ultimately deposited at the surface, the removal process is largely influenced by the sub-grid model. However, the sub-grid model that determines the sensitivity of smaller-scale transport explicitly does not encapsulate all deposition processes (such as local coalescence, surface waves, and smaller unresolved-scales of turbulence). The question whether particle deposition is more of an artificial manifestation of the sub-grid modeling suggests that an additional sub-grid sensitivity test can retroactively inform the results of this study.

Provided that the sub-grid model does not cause substantial deviation from the overall particle transport, the purpose of this work is to inform field observation practices by presenting baseline characteristics of ensemble Lagrangian particle trajectories. Shear- and buoyancy-driven turbulence that may give rise to coherent structures dictate the manner of Lagrangian trajectories and thus the concentration distribution and residence time of aerosol particles, shown in Sections 3 and 4.3. This residence time modification due to stability regimes can then be utilized alongside field observations by setting expectations regarding the locality of aerosol particles.

Extensions of this work must include additional vertical forcing due to clouds (e.g., note the cloud streets in Figure 1(b)), and consequently the particle hygroscopicity and a host of transient phenomena. Nevertheless, this work points to many



considerations for aerosol observation and the inherent sub grid scale parametrization and grey zone problems faced when
attempting to close observations with model simulations. First, when measuring near a specific surface emissions event in even the most slightly unstable environment, there is a critical time/distance when the plume is entrained into a narrow but strong updraft and quickly transitions from the near surface environment to near the top of the mixed layer. This transition is over a narrow spatial scale and may be difficult to observe and even more difficult to link observations to models to infer fluxes. Spanwise observations may miss this critical point, as the absolute distance from the source and the vertical
velocities are unknown. Likewise, in the streamwise direction, much depends on if the aircraft is passing in a narrow strong updraft or wider and weaker downdraft; but spatially the odds are that a weaker downdraft is observed (e.g., Figure 3). Even for neutral conditions, it can take over 90 minutes of transport in our simulations before a significant number of particles reach the top of the mixed layer, and even then, the center of the plume is in mid-mixed layer. The nature of such features is demonstrated during measurement of offshore flow conditions, such as Reid et al. (2001). Two cases in particular were
highlighted at 8 and 12 m/s wind speed. Interestingly, the lower wind speed condition exhibited higher particle concentrations than the higher - not necessarily because of a larger particle production, but simply due to decreased stability and increased diffusion for the higher wind speed case, including the formation of turrets.

A second implication of these simulations is the demonstration of differences in lifetime as a function of altitude. Furthering this work in the context of particle age, surface scavenging and cloud processing is a topic of an ongoing effort.
Nevertheless, the particle trajectories generated by the LES used here give insight on particle evolutionary processes. To be sure, particles near the surface source statistically are younger than particles at the top of the MABL. However, this work demonstrates that any given parcel hosts a distribution of ages. Some particles can remain near the surface for some time (especially for neutral and expectantly stable conditions), and some particles can be very quickly transported upwards quite rapidly, even in neutral conditions. The ensemble of simulations in Figure 8 show a persistent enhancement at the inversion;
some updrafts puncture the inversion layer, and detrain particles to the stable layer where they remain at that altitude for over an order of magnitude longer than particles that are just a few 10s of meters below them. In a more realistic MABL simulation with clouds, further exchanges will take place between the MABL and free troposphere disrupting such a long lifespan at that level. But the point still remains, significant differences in the distribution of particle lifetimes as a function of altitude may need to be considered in some experiment designs.

**6.0 Data availability**

Model data for this project can be found at the following website:
 https://curate.nd.edu/projects/Lagrangian_aerosol_particle_trajectories_in_a_cloud-
free_marine_atmospheric_boundary_layer_Implications_for_sampling/226284
SAR data for Figure 1 is provided at https://www.star.nesdis.noaa.gov/socd/mecb/sar/sarwinds_rcm_rs2.php



7.0 Acknowledgements

The authors would like to thank Coda Phillips of the University of Wisconsin for guidance in writing post-processing code, Chris Jackson of NOAA STAR for generating the SAR image used in Fig. 1, and Emily Melvin of the University of Notre Dame for helpful discussions in the development of this study.

8 Financial Support

Support for HP was provided through the National Research Council: Research Associateship Programs under the Greater Laboratory Science Interchange Program (GLSIP) by the NRL Director of Research and NASA. JR was supported by the NRL Base Program and ONR 322. DHR was supported by ONR Code 322 Grant N00014-24-1-2765. Authors CJ and MC were supported by ONR 322 under the MAGPIE campaign. This work was supported in part by a grant of computer time from the DoD High Performance Computing Modernization Program at Onyx. Resources supporting this work were provided by the NASA High-End Computing (HEC) Program through the NASA Center for Climate Simulation (NCCS) at the Goddard Space Flight Center as part of the NASA Clouds, Aerosol, and Monsoon Processes-Philippines Experiment (CAMP$^2$Ex).

9 Author contributions

HP performed the core modelling simulations, created all figures but Fig., 1,  and drafted early portions of the paper. PC performed the MARBLES simulations JR and DR contributed significantly to later drafts of the paper. MC performed reviews and revisions.

10 Competing Interests

The contact author has declared that none of the authors has any competing interests.

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
