# Peer review of "Lagrangian aerosol particle trajectories in a cloud free marine atmospheric boundary layer: Implications for sampling"

_EGUsphere, 2025_

## Author Comment (AC1)

**Reviewer 1:**

The authors thank the reviewers for their comments. We hope to have addressed all their concerns.

This paper was a pleasure to read. The results are not particularly profound or unexpected, but are a very nice presentation of how to think about particle trajectories and histories. The writing is clear and straightforward and the graphics are generally well chosen to get the points across. Most of my thoughts while reading were about how nice it would have been to extend the work farther rather than about whether I trusted the reasoning. I don't actually have a lot of substantive comments.

I was a bit confused by figure 6. The legend describes panels b through d as "the time-height evolution of the column count normalized deposited particle concentration" whereas the text described those plots as "the probability of a deposited particle's maximum altitude reached, given a particular lifetime". I don't understand the first phrasing at all, but the second makes some sense to me.

Response: We agree with the confusing description of Figure 6. We have modified the figure caption to "the probability of a particle's maximum altitude achieved given a particular lifetime".

I was surprised not to see explicit mention of an apparent pattern that the slightly unstable plots would look a lot like the unstable ones if the x-axis were $T_{eddy}$. Yes, Figure 8 shows that some details would differ, but the overall impression is that characterizing mixing in the MBL simply with $T_{eddy}$ would be useful.

Response: We have added a few explicit references to the fact that certain features occur at similar values of $T_{eddy}$, including in the discussion of Figures 6 and 8.

I'm trying to think of whom this information would be really valuable. After all, a sudden release of low-altitude 10 μm particles is not a common occurrence! I wind up thinking about time scales of mixing: how long after a front passage or scavenging event (a rain storm) do I have to wait before I can assume that aerosol in the mixed layer is well-mixed? How often is a particle likely to have encountered clouds in stratocumulus or trade wind cumulus regimes? What do I need to know to make those estimations? These questions would be relevant to sampling expeditions or modeling.

Response: This is an excellent point, and one of our primary motivations. We chose a pulse release to not simply replicate a sudden event: rather, it helps us address precisely the questions you pose regarding the mixing time. We could certainly apply a continuous

release at the lower surface (and indeed we have in other simulations), but mathematically our results would be unchanged since we are looking at statistical properties of individual particles over their lifetime. That is, if we applied a continuous release but shifted the particle properties to be relative to when they were released, we could regenerate every figure in this manuscript. Thus one interpretation of our "pulse" is that it is doing this time shift automatically. As the authors point out, in the real world we'd like to know how long the particles have taken to get where they are, *relative to their own start time*, and this allows us to do precisely that.

We have added a brief explanation in this regard in the manuscript where the particle injection is described:

"We emphasize that while the particles in the simulations are released as a single pulse, this is not meant to literally represent such an event, which is a rather unrealistic situation. Rather, this technique allows us to automatically reference the individual particle trajectories and lifetimes to a common reference point; i.e., the statistics presented below could be exactly generated with a continuous release of particles at the surface, where Lagrangian statistics are computed relative to an individual particle's generation. So while in the real MABL newly generated aerosol particles would be continuously injected into a populated background, this technique allows us to speak directly to the individual fate of a single new particle and distinguish its lifetime and position relative to other nearby particles. This has strong implications for Eulerian-based sampling strategies and interpretations of particle observations, including assumptions made about their exposure time and distance from their source."

Line 163--4: "The particle sizes are set to 10μm in aerodynamic diameter to represent coarse mode particles, which is much smaller than the smallest turbulence scales of the flow" Well, yes, but any realistic particle size is smaller than the turbulence scale. I expect you're referring to stopping distance for the particle being much smaller than turbulence scales or terminal velocity much less than typical vertical winds.

Response: Yes, though some large droplets (spray, rain, etc.) could easily exceed the smallest turbulence scales and have appreciable inertial effects. We are trying to differentiate from these cases. We have added a parenthetical comment "unlike large droplets including rain and spray" to this line.

Equations 2 and 3: Inconsistent use of boldface to indicate vector quantities

Response: Thank you for noting the inconsistent use of boldface, we have modified the text accordingly.

Line 210: Is $Q_*$ supposed to be $Q_0$?

Response: Yes, the correct expression is Q0 instead of Q*, we have modified the text accordingly.

Line 283: "There is a slight crossover in wind speeds at the top of the mixed layer, with neutral having a lower wind speed at 9.2 m/s, and unstable at 10.7 m/s" That crossover is at something like 30 meters, hardly the top of the mixed layer. Seems to be more at the transition between a surface layer and the bottom of the central mixed layer.

Response: Yes, this was a typo and was meant to refer to the "top of the surface layer".

The lower x-axis in Figure 2b is paradoxical. A log scale can't go to zero. Is it linear between $-10_{-4}$ and $+10_{-4}$? That would explain the kinks in the blue and orange lines and the smooth passage through 0. Makes it hard to imagine dividing the blue lines by 10. Not sure I know of a better way to present the data though.

Response: The reviewer is correct: we are using a signed-log-scale outwards of $10^{-4}$, and between we are using a linear scale. We are trying to simultaneously highlight the features of the profile (which require a log scale to see), but also the inherently signed nature of the TKE production terms. We realize now that we had never specified this, and indeed having "0" on the x-axis is confusing. We have provided an explanation in the caption.

Figure 3: It is gorgeous, but since the w' color scale is biased, it looks like there are net downdrafts since a 0.4 m/s downdraft looks just as saturated as a 0.8 m/s updraft. Does it not work with a symmetric color scale, leaving the strongest downdrafts unsaturated?

Response: The purpose of the scale being set to an asymmetrical color scale but with white at zero is to emphasize the state of less frequent, but stronger values of positive w' (updrafts), in contrast to the more frequent, but smaller values of negative w' (downdrafts). We have added an additional note on the Figure description. These features are difficult to see when the color scale is symmetric. We have provided this explanation in the caption.

Figure 6a: It appears that the most probable lifetime is much shorter than the 1000 s you mention. If the data are saved every 5 s, you could have shown even shorter periods at the beginning of the run. Does that not work?

Response: Unfortunately, the information to make Figure 6a is not included in the particle statistics that are written every 5s. To generate Figure 6a, we had to run an additional set of simulations past $10^5$ seconds (in order to get the full lifetimes). However this required us to decrease the frequency of statistical output due to computational storage requirements.

I'd be interested to see something like 6a, but with fraction of original particles remaining. It wouldn't have the nice dips in the unstable cases that you point out, but a flattening of the curve, so it wouldn't be as striking a plot, but would be easier to understand.

Response: The attached figure illustrates the fraction of original particles remaining for the first 6 hours of the run (approximately one e-fold loss)

[Figure]

Indeed, it has the features which the reviewer predicted. However, since our primary purpose of Figure 6a was the emergence of the bimodal lifetime distribution, we have chosen to keep the original version since this is not as clear on the above figure. However, we have added the above figure to emphasize the reviewer's point.

Line 593: "spatially" is missing the y

Response: Thank you, we have corrected the text.

Line 627: "Even for neutral conditions, it can take over 90 minutes" implies that unstable conditions take longer! Perhaps just ditch the "Even"

Response: Thank you, we have removed the word "Even".

---

## Author Comment (AC2)

**Reviewer 2:**

Overall: This is an interesting study with valuable insights into particle motion throughout the MABL. It points out many relevant and meaningful gaps in existing SSA research related to the transport of these giant particles from the surface throughout the MABL. I think it represents novel research that deserves to be published. Below, I've posted a few questions to consider that I think could be meaningful discussions/alterations to the manuscript.

Methods:
I understand the method releases a singular aerosol plume event to observe the evolution of these particles throughout the LES, however, I wonder if you continually released particles and allowed them to loop through the periodic boundary conditions if the mixing eventually reaches some sort of homogeneity? In the intro, you preface that this research could impact/provide insights into proposed in situ sampling strategies and that something like cloud streets might have some impact on where whitecaps are being generated. While SSA are produced by these plumes, they exist within a background concentration of previously produced aerosol. If you were to continually release these particles throughout the whole domain (or say, at a length scale of every whole number of $T_{eddy}/T_{neut}$ to represent increased production at these eddy lengths), how prevalent are these "near-absence of particles near the surface" (line 387)? Is the focus on the lowest surface layer in this statement? If so, I believe this can be made more clear.

This is an excellent point, and centers on a point which is one of the central themes of this work. The homogeneity the reviewer speaks of is very relevant to the Eulerian-based measurements that one might make. Indeed, newly generated particles would get mixed into the already-laden MABL.

However, the question is: if you sample particles at some height in a well-mixed MABL, how long have they been there? Which have been recently released? In that sense, the "near-absence of particles" refers to the near-absence of *newly created* particles (even if there are plenty of others which have been sampled). It is this Lagrangian perspective that we are trying to understand. We have added a brief explanation in this regard in the manuscript where the particle injection is described:

"We emphasize that while the particles in the simulations are released as a single pulse, this is not meant to literally represent such an event, which is a rather unrealistic situation. Rather, this technique allows us to automatically reference the individual particle trajectories and lifetimes to a common reference point; i.e., the statistics presented below could be exactly generated with a continuous release of particles at the surface, where Lagrangian

statistics are computed relative to an individual particle's generation. So while in the real MABL newly generated aerosol particles would be continuously injected into a populated background, this technique allows us to speak directly to the individual fate of a single new particle and distinguish its lifetime and position relative to other nearby particles. This has strong implications for Eulerian-based sampling strategies and interpretations of particle observations, including assumptions made about their exposure time and distance from their source."

An additional question I have is what percentage of these particles exist within the primary (trapped at the surface) vs. the secondary (entrained into the MABL) modes? If you have statistics on this, those are really novel and particularly interesting in understanding how many of these giant particles are functionally being entrained.

Response: We thank the reviewer for this insightful question regarding the partitioning between the two transport modes. Indeed, this quantification provides important insights on coarse mode aerosol entrainment efficiency. To address this, we analyzed our particle trajectory data using the following criterion: particles are classified as "first mode" if their maximum altitude never exceeds 100m (approximately the surface layer depth), while "second mode" particles are those that reach above 100m and thus experience MABL-scale circulation. The 100m threshold was chosen based on inspection of Figures 5 and 6, where the transition between surface-trapped and vertically-transported particles becomes evident. While we acknowledge the somewhat arbitrary nature of this threshold, it provides a practical metric for quantifying the transport mode partitioning, and a more sophisticated classification metric is beyond the scope of the present study.

Based on this analysis, the mode partitioning is as follows:
- Neutral conditions: 26.0% first mode, 74.0% second mode
- Slightly unstable: 26.4% first mode, 73.6% second mode
- Unstable: 21.7% first mode, 78.3% second mode

We have added these statistics to Section 3.3 (line X), following the discussion of Figure 6: "To quantify this bimodal transport behavior, we classified particles based on whether their maximum altitude exceeded 100m. This threshold was chosen based on visual inspection of Figures 5 and 6, which suggest that around this height there is a transition point where particles either remain trapped near the surface or become entrained into the mixed layer circulation. This analysis reveals that 26.0%, 26.4%, and 21.7% of particles never escape above 100m in neutral, slightly unstable, and unstable conditions respectively, confirming that instability fundamentally alters the balance between surface-trapped and MABL-entrained particle populations. The complementary fractions (74.0%, 73.6%, and 78.3%) that do reach the mixed layer correspond to the second mode particles whose extended lifetimes arise from their participation in MABL-scale circulation."

Section 2.2: What is the equivalent dry particle diameter/radius for this 10 micron aerodynamic diameter particle? Is it dry? If it's dry, a small discussion on how these giant SSA may act when their particles are deliquesced (say at different altitudes, with differing RH, within the MABL) would be interesting. These particles will grow and shrink through these many eddy cycles. How applicable are your PDFs of residence times at different regions in the MABL (Fig. 7) when these particles experience dynamic fall velocities as a function of changing size due to RH changes? Also, if your particles are dry, what is the equivalent sized particle in situ that this 10 um aerodynamic diameter size represent for an SSA particle at formation diameter (d_99), which is ~4x the diameter with a mass that is mostly water (~half the density of salt)? This has huge implications for in situ scientists trying to gauge which size this is relevant for. A quick back of the envelope estimate is that the particles in this study are likely representative of a < 3 um particle at formation diameter, which after undergoing evaporation outside the surface layer, become exceptionally well mixed throughout the MABL and aren't subject to much removal through dry deposition.

Response: This is an extremely good point, and one which is the subject of ongoing research. For the purposes of this manuscript, we avoid making the distinction between dry/wet diameter because we are neglecting swell/evaporation/condensation. Functionally the settling velocity is so small relative to eddy diffusivity that settling velocity is not a significant consideration for the work being demonstrated here. But everything the reviewer points out is relevant, especially as particles become close to the oceans surface or cloud base as discussed in the previous comment. The next set of simulations are currently being mapped out to generate similar statistics but with realistic humidity fields at the boundaries were the droplet size may start to be relevant.

Lines 346-350: Yes, the particles in the unstable case go through more cyclic motions in the 6000 s time, but isn't this just a function of the eddy roll lengths determined by the MABL stability? Say, if you were to generate the same graph as Figure 4 but with a normalized X-axis to T_eddy/T_neut, with Time (s) varying on the upper X-axis, would these graphs look much different from each other? Time is a valid means of presenting the research, but I wonder how different these results would look if presented from an eddy roll length perspective. For example, Figure 5 demonstrates the same " aerosol at approximately 2.5 T_eddy/T_neut for all three conditions (neutral, slightly unstable, and unstable) (highlighted).

Response: This is a good point, but only for a sufficiently strong unstable stratification. For decreasing stability, the definition of an "eddy roll length" loses meaning. As shown in Figure 3 of the manuscript, the w' experience much less coherence as the stability approaches neutral. With this in mind, consider the figure below (Figure 4 of the manuscript but with the upper X-axis fixed to 2 eddy roll lengths depending on stability):

[Figure]

We see for (c) the particles' strong vertical transport due to the presence of coherent roll structures. This, in turn, leads to a state of an absence of particles near the surface most visually apparent in Fig 5(c). For the neutral configuration, there would be no such scenario however long of time the simulation progresses due to lower levels of vertical transport that lack vertical velocity coherence.

With appreciable instability, we do expect the motions to scale with $T_{eddy}$, and we have added comments accordingly in our discussion of Figures 6 and 8.

Overall Writing and Formatting: The writing is really great, which is interesting, because it's in direct contradiction to all of the copy-editing that needs to be done on this manuscript. The lack of attention to minor details in the writing as well as figures generates detracts from the paper. The formatting of the many unit notations is inconsistent across the paper (italicized, non-italicized, inconsistent spacing). I will try to provide some proposed copy-edits and inconsistencies I saw in the attached PDF, but ultimately this manuscript should

be copy-edited professionally before any future submission.  Additionally, the citation formatting is all over the place - I can't tell if these have been manually input or why there's such inconsistencies, but these need to be non-italicized and formatted properly, linking to their citations at the end of the paper. There's many things that need to be improved to comply with Copernicus publishing standards, including colorblind friendly figures and increasing the dpi of some figures (why does Figure 5 have a distorted aspect ratio?) - https://publications.copernicus.org/for_authors/manuscript_preparation.html#figurestables

Response:
We have gone through the manuscript with respect to the citation formatting and fixed minor details in the overall writing and figures. We hope to have addressed the reviewer's concern.

---

## Author Response (AR2)

Hi, As per discussion with the editor, no corrections were made. We should not make text changes associated with figure 3, and the 'cutoff" was a pdf artifact caused by using the "make a pdf" feature versus using the printer module.
Thanks
Jeff

Dear Jeff,
thank you for your message.
The comments were given by the reviewer and I simply copied them.
In the pdf manuscript file the y axis labels are indeed cut, so please make sure that in the final figure version everything is correct.
Regarding re-organizing Figure 3: I agree that changing the figure would require changing the text, but rewriting the text is not an option anymore for an accepted version, so please leave the order of Figure 3 unchanged.
Best regards,
Johannes

Am 16.07.2025 um 10:10 schrieb Vitaly Muravyev:
Dear Johannes Schneider,

Please consult Jeff's message below and advise them on how to proceed.

Kind regards,

Vitaly Muravyev

[Figure]

Vitaly Muravyev
Editorial Support

Copernicus GmbH
Bahnhofsallee 1e
37081 Göttingen
Germany

Phone: +49 551 90 03 39 47

Copernicus GmbH
USt-IdNr.: DE216566440
Based in Göttingen, Germany
Registered in HRB 131 298
County Court Göttingen
Managing Director Thies Martin
Rasmussen

Data protection remark: emails sent by us might include the email address
editor@mailarchive.copernicus.org as CC. When replying to such emails and keeping this
email address in CC, your replies will be archived in the online system CO Editor alongside
the manuscript identified through the manuscript number in the subject line. Such
archived emails are accessible for the respective handling editors, journal's chief editors,
as well as Copernicus Publications' staff members.
* * *
From: Reid, Jeffrey S CIV USN NRL DET MONTEREY CA (USA)
Sent: Monday, July 14, 2025 16:28
To: Copernicus Publications Editorial Support
Cc: david.richter.26@nd.edu; editor@mailarchive.copernicus.org
Subject: RE: egusphere-2025-576 (author) - manuscript accepted with corrections

Hi Johannes,
Thank you for the provisional acceptance. Question for the editors. The
technical correction requested listed is "The only suggestion I have is
about Fig. 3, which might be more intuitive if the higher altitudes were
placed above lower altitudes. Also, the y labels in the 300 m/slightly
unstable panel are cut off." We went back to the postscript file of the
figure, and there is no cutoff y axis-this may be a pdf thing. Second, if
you so desire, we can reorganize figure 3, but in doing so we would have to
rewrite a page of text. The discussion starts at the bottom of the MABL and
then works its way up, which from a forcing physics point of view makes the
most sense. But please let us know if you wish a rewrite.
Be well,
Jeff
* * *
Jeffrey S. Reid, Ph.D. Code 7543
U.S. Naval Research Laboratory | phone: (831) 262-9052
jeffrey.s.reid20.civ@us.navy.mil

"My logisticians are a humorless lot ... they know if my campaign fails, they are the first ones I will slay." - Alexander the Great